# Reducing Calf Mortality in Ethiopia

**DOI:** 10.3390/ani12162126

**Published:** 2022-08-19

**Authors:** Johanna T. Wong, Jennifer K. Lane, Fiona K. Allan, Gema Vidal, Ciara Vance, Meritxell Donadeu, Wendi Jackson, Veronica Nwankpa, Shubisa Abera, Getnet Abie Mekonnen, Nigatu Kebede, Berhanu Admassu, Kassaw Amssalu, Alemayehu Lemma, Tsegaw Fentie, Woutrina Smith, Andrew R. Peters

**Affiliations:** 1Supporting Evidence Based Interventions-Livestock, Royal (Dick) School of Veterinary Studies, University of Edinburgh, Easter Bush Campus, Midlothian EH25 9RG, UK; 2One Health Institute, School of Veterinary Medicine, University of California, Davis, CA 95616, USA; 3Faculty of Veterinary and Agricultural Sciences, University of Melbourne, Werribee, VIC 3030, Australia; 4Department of Microbial Cellular and Molecular Biology, Addis Ababa University, Addis Ababa P.O. Box 1176, Ethiopia; 5Animal Health Institute (AHI), Sebeta P.O. Box 04, Ethiopia; 6Aklilu Lemma Institute of Pathobiology, Addis Ababa University, Addis Ababa P.O. Box 1176, Ethiopia; 7Feinstein International Center, Tufts University, Boston, MA 02111, USA; 8Ministry of Agriculture, Addis Ababa P.O. Box 62347, Ethiopia; 9College of Veterinary Medicine and Agriculture, Addis Ababa University, Bishoftu P.O. Box 34, Ethiopia; 10College of Veterinary Medicine and Animal Sciences, Tewodros Campus, University of Gondar, Gondar P.O. Box 169, Ethiopia

**Keywords:** calf mortality, youngstock mortality, peri-urban production, pastoral production, mixed crop–livestock production, diarrhea, respiratory disease, Ethiopia

## Abstract

**Simple Summary:**

Disease and death of young livestock cause financial and production difficulties to farmers around the world. High rates of disease and death occur in various production systems in Ethiopia, hampering livestock production, reducing incomes, and damaging livelihoods. Over the last 10 years, studies carried out in Ethiopia have reported death and disease incidence rates in young livestock as high as 31% and 67%, respectively. Diarrhea and respiratory infections are the two leading causes of disease and death in calves in all production systems. In this paper, we describe findings from the experience of the Young Stock Mortality Reduction Consortium. This unique group produced important information on the main causes of disease and death in Ethiopia and created activities for small-scale farmers to address these problems. We found that several diseases caused diarrhea and respiratory infections in young calves in Ethiopia. Improving farmer knowledge and behaviors with respect to basic livestock management led to considerable reductions in young livestock disease and death and has the potential to help improve livestock productivity and human livelihoods in Ethiopia.

**Abstract:**

Morbidity and mortality of young stock present economic and production challenges to livestock producers globally. In Ethiopia, calf morbidity and mortality rates, particularly due to diarrhea and respiratory disease, are high, limiting production, incomes, and the ability of farmers to improve their livelihoods. In this paper, we present findings from the combined experience of the Young Stock Mortality Reduction Consortium, which conducted epidemiological and intervention testing in calves across three production systems. This innovative alliance identified *Cryptosporidium parvum* and *E. Coli* K99 as the most common causes of diarrhea in pastoral and peri-urban calves; *Strongyloides* spp. as the most common fecal parasite in mixed crop–livestock and peri-urban calves; and bovine adenovirus, parainfluenza virus-3, and bovine respiratory syncytial virus as the most common respiratory pathogens in peri-urban calves. Furthermore, by improving producer knowledge with respect to fundamental livestock husbandry, feeding, housing, and neonatal care practices, calf mortality risk across production systems was reduced by 31.4 to 71.4% compared to baseline (between 10.5 and 32.1%), whereas risk of diarrhea was reduced by 52.6–75.3% (baseline between 11.4 and 30.4%) and risk of respiratory disease was reduced by 23.6–80.8% (baseline between 3.3 and 16.3%). These findings have informed scaling strategies and can potentially contribute to improved livestock productivity and human livelihoods in Ethiopia.

## 1. Introduction

Ethiopia has Africa’s largest livestock populations, contributing to approximately 45% of the country’s total value of agricultural production [1,2]. Of the livestock species, cattle are considered the most economically significant [1,3]. In 2019, the cattle population in Ethiopia was estimated to be 70 million head, comprising almost all local breeds, with most raised in small herds of one to nine head [4]. Five production systems are used: 76% of cattle are kept in mixed crop–livestock systems, 14% in pastoral/agropastoral systems, 7% in urban/peri-urban systems, 2.5% in commercial dairies, and 0.5% in feedlots [2]. Overall, the majority (60%) of cattle are 3–10 years of age and are mainly used for breeding (25%) as replacement stock, draught power (22%), and milk (11%) [4].

Income from livestock accounts for 11% of total income in rural households [5], and livestock development is considered to be fundamental to the sustainable growth and transformation of Ethiopia [2,6]. However, in the last decade, the majority of the growth in the cattle subsector in Ethiopia has been due to increased numbers of animals or farmers rather than improvements in productivity [3]. To improve the sustainability of livestock growth in Ethiopia, a shift to improved productivity is required.

High rates of morbidity and mortality, particularly in young stock, have constrained progress and limited growth of the cattle sector [3,6]. Small-scale studies conducted within the last ten years report dairy calf mortality incidence risks of between 12 and 20% in urban or peri-urban systems [7,8,9] and 31% in mixed crop–livestock systems [10]. Where reported, calf morbidity incidence risks are higher, ranging between 34 and 67% [8,10,11]. Diarrhea is the most common morbidity reported, followed by respiratory disease and navel ill, with poor colostrum consumption, nutrition, and hygiene management being common risk factors [7,8,9,10,11]. Among the available literature, more research has been conducted within the urban/peri-urban/intensive dairy farming sector (including cross-bred cattle) than mixed-crop and pastoral systems, despite the latter two comprising the majority of farms in Ethiopia. Scarce information is available for pastoralist systems.

High mortality leads to not only a loss of income but also a loss of replacement stock and genetic material, making it difficult for farmers to replace their losses or expand their herds [3,9]. Environmentally, high mortality rates also contribute to the waste of finite resources and increased emission intensity due to decreased efficiency [12]. The general recommendations for farmers to improve calf colostrum intake, provide higher quality feed, improve environmental hygiene, and seek veterinary services in order to reduce calf morbidity and mortality are repeated throughout the literature, indicating the persistence of these issues over time and across production systems in Ethiopia.

To contribute to resolving some of these issues, the Young Stock Mortality Reduction Consortium (YSMRC) was formed under the auspices of the Ethiopian Ministry of Agriculture (MoA). Members of the YSMRC have included Addis Ababa University’s College of Veterinary Medicine and Agriculture (AAU-CVMA); Aklilu Lemma Institute of Pathobiology (AAU-ALIPB); the University of Gondar; the National Animal Health Diagnostic and Investigation Centre (NAHDIC); Tufts University; Supporting Evidence-Based Interventions in Livestock (SEBI-Livestock); and the School of Veterinary Medicine, University of California, Davis. An innovative and novel funding mechanism made this project possible, with USAID Feed the Future Innovation Lab for Livestock Systems, the Bill & Melinda Gates Foundation, and the Ethiopian Ministry of Livestock and Fisheries all providing financial support for the project. Through this collaboration, a pilot project was carried out to generate key epidemiological information on the major causes of youngstock morbidity and mortality that hampers the potential productivity of livestock in Ethiopia and to identify and to assess the impact of a package of basic health and husbandry interventions on youngstock mortality within different production systems and ecological zones in Ethiopia. Whereas disease surveillance was conducted in cattle and small ruminants, and health and husbandry interventions were piloted for cattle, small ruminants, and camels, in this paper, we only present the YSMRC findings with respect to cattle findings.

## 2. Materials and Methods

### 2.1. Study Design

Initially envisioned as a longitudinal study with initial disease surveillance informing the design of the interventions, due to implementation challenges and time constraints, we concurrently conducted disease surveillance activities with an intervention design and implementation. In this manuscript, the two arms of the project are referred to as the epidemiological arm, which comprised the disease surveillance portion, and the intervention arm, which refers to the implementation of the interventions. Households for the two study arms were recruited from the same study areas.

The YSMRC was launched in 2016; field work activities for the epidemiological arm were conducted from November 2017 to August 2019. Standard operating procedures (SOPs) to guide the intervention arm were collaboratively developed and refined from 2017 to 2018, with staggered baseline evaluations conducted across different study areas from March to August 2019, prior to the introduction of the interventions. Year-long interventions were implemented following baseline data collection, and the staggered final evaluations ran from March 2020 to July 2020, timed to match the baseline evaluation as best possible to limit seasonal variations. Questionnaires and all SOPs developed for the interventions are available in Appendix A.

There is substantial overlap in the households and animals enrolled in the epidemiological and intervention arms of the project; however, due to differences in household identification, anonymization, and data management strategies, we were unable to merge the databases for the two arms of the study, and the results are presented under separate subheadings.

#### 2.1.1. Study Area

Study regions representing three major livestock production systems in Ethiopia were selected in consultation with livestock health extension officers from the Ministry of Agriculture (MoA). Selection criteria for the study sites included prevailing production system, species of animal, livestock population density in the area, and accessibility.

Six study districts were selected across five regions of Ethiopia (Table 1 and Figure 1).

#### 2.1.2. Household Selection

Eligible farms and households were identified after reviewing regional livestock office registers. Following identification of eligible households, selection criteria for both arms included that households owned at least one of the following: pregnant heifers/cows or cows with a calf (<6 months of age); a willingness to participate throughout the study period; and, preferably, that the household had previous experience working with livestock extension officers. For the epidemiological arm, young livestock (<6 months old) were convenience-sampled in a cross-sectional study design. Intervention-arm households were convenience-sampled and followed longitudinally over a one-year period.

### 2.2. Epidemiological Arm

#### 2.2.1. Sample Size and Methodology

A comprehensive study evaluating multiple disease pathogens in multiple species and production systems across Ethiopia had not been conducted at the time of study design, and disease prevalence estimates for pathogens in Ethiopia vary widely, with large variances between the prevalence reported for the same pathogens. For the epidemiological arm, disease prevalence and livestock populations were estimated based on smaller studies in single production systems or regions in Ethiopia and triangulated with Ethiopian livestock census data. Sample size requirements were determined using these estimates, with an alpha of 0.05 and specificity of 95% [8,13,14,15,16]. Furthermore, identify risk factors at a level of confidence of 95% (alpha = 0.05), a power of 0.80 and a sample size of at least 1000 enrolled animals were calculated, reflecting different farm sizes and variation between production systems and to account for loss to follow-up.

#### 2.2.2. Data Collection

Standardized data collection forms (Appendix A) were utilized to collect farm and individual animal-level information for both demographic information and risk factor analysis. A physical exam was performed on all calves enrolled in the study. Standardized clinical scores based on the University of Wisconsin–Madison School of Veterinary Medicine Calf Health Scoring System were used to characterize and quantify findings, including cough, ear position, ocular and nasal discharge, and fecal consistency (Appendix A). Standard sample collection protocols were followed to obtain fecal samples, respiratory swabs, and whole blood from eligible calves less than six months old. Animals enrolled and tested included both healthy and clinically affected animals. Due to logistical, laboratory, resource, and other limitations, diagnostic assays were conducted on subsets of the collected samples. Neonatal diarrhea complex pathogens, including *E. coli* K99, *Cryptosporidium parvum*, bovine rotavirus, and bovine coronavirus (CoV) were tested for using a combination of diagnostic assays, including the commercially available Pathasure Enteritis 4 antigen ELISA kit (Biovet, Saint-Hyacinthe, QC, Canada; https://www.biovet-inc.com/en/product/pathasure-enteritis-4/, accessed on 31 May 2022), as well as traditional microbiological bacterial culture and antibiotic sensitivity testing. Traditional fecal flotation technique and microscopic examination were used to screen fecal samples for evidence of fecal parasite infection in calves from mixed crop–livestock and peri-urban farms. Respiratory pathogens, including bovine respiratory syncytial virus (BRSV), parainfluenza virus type 3, bovine adenovirus, bovine herpesvirus-1 (BHV-1), infectious bovine rhinotracheitis (IBR), *Mannheimia haemolytica,* and *Pasteurella multocida* were tested for using a combination of diagnostic assays, including IDEXX serological assays, Trivalent Ab test (no longer commercially available), BHV/IBR gB X3 Ab test (IDEXX, Westbrook, ME, USA; https://www.idexx.com/en/livestock/livestock-tests/ruminant-tests/idexx-ibr-gb-x3-ab-test/, accessed on 31 May 2022), and traditional microbiological bacterial culture and sensitivity testing. Passive transfer of immunoglobulins status was assessed using a commercially available radial immunodiffusion (RID) test kit (Kent Laboratories, Bellingham, WA, USA; https://kentlabs.com/jjj/product/bovine-igg-test-kit-radial-immunodiffusion-test-kit/, accessed on 31 May 2022), considered the gold standard for testing.

For the epidemiological arm, the presented analyses were predominantly focused on descriptive results. Categorical variables were summarized as counts and percentages. Continuous variables were summarized with the mean and standard deviation (SD). Associations between categorical variables and other study variables, such as production system, were analyzed using chi-Square tests. In cases in which the sample size was small, Fisher’s exact test was used instead. Associations between continuous variables and production system were analyzed using ANOVA. Associations between continuous variables were explored by using linear regression and locally estimated scatterplot smoothing (LOESS) models. All variables were initially screened, and results with a *p*-value ≤ 0.2 triggered further inquiry. Where statistical interpretations are provided, a significance level of a *p* ≤ 0.05 suggests a strong association between study variable(s) and the outcome of interest.

### 2.3. Intervention Arm

#### 2.3.1. Sample Size and Methodology

The main objective of this arm was to pilot the interventions for applicability and affordability. Within each of the six districts noted earlier, three kebeles/wards were selected per district (kebeles generally have three villages, each with 150 households). Within each kebele, one village was purposively selected, with 50 households from that village then randomly selected, resulting in 150 households/district. To avoid substantial differences in traditional practices during the implementation phase, neighboring kebeles within a district were selected. In total, 900 households were invited to enroll in the study, representing 18 villages from 18 kebeles in six districts. A post hoc power analysis of the calf mortality mixed model was performed using the simr package in R (Version 4.0.3, The R Foundation for Statistical Computing, Vienna, Austria), which showed that with an alpha of 0.05, the model had 100% power (95% confidence interval: 69.15–100%).

#### 2.3.2. Interventions

In collaboration with local consultants, the YSMRC created an extensive list of interventions for each production system and developed SOPs for all interventions. However, distributing the SOPs and training producers in all interventions was not feasible. During a validation workshop and a consultative meeting with SEBI, selection of key interventions was suggested. For each production system, interventions were selected through consultation with stakeholders and experts, aiming for targeted improvement in animal husbandry, management, and health. The full list of interventions that farmers received training on is included in Appendix B (Table A1).

A monitoring and evaluation plan was developed, including a results framework, through which indicators were selected to monitor intervention uptake and changes in key indicators. Production system characteristics and feasibility of data collection were among the considerations with respect to the selection of monitoring parameters, and study questionnaires were pretested before use. Data such as reproductive parameters, including birth and death of young stock; health outcomes, including incidence of diarrhea and respiratory disease; and intervention uptake (Table 2) were collected at baseline and final evaluations.

The interventions were introduced to participating farms via training sessions. Participating households were then continuously coached by extension officers throughout the study period.

#### 2.3.3. Data Collection

Data were collected using questionnaires completed by trained enumerators. Bespoke software was used to standardize data entry from the questionnaires, and data were stored on the Bases & Datos server. This application was created by Iñaki Albizu (Zaragoza, Spain) using FileMaker Pro software (Version 12, Claris International Inc., Cupertino, CA, USA), as the database engine. Data were then exported to Microsoft Excel (Version 2013, Microsoft, Redmond, WA, USA), where they were cleaned. Statistical summaries were produced using Excel and R (Version 4.0.3) via RStudio (Version 1.3.1093, RStudio, Boston, MA, USA).

Changes in practice in the intervention area between baseline and final evaluation were assessed. Households were evaluated as having (1) made an improvement to practices in the intervention area; (2) made no change but were already performing the recommended practice; or (3) made no change and were not performing the intervention as recommended or had a negative change in the intervention area.

For each enrolled farm, mortality risk was calculated for the previous 12 months and defined as the number of live-born calves that died divided by the total number of calves born alive as a percentage. Average mortality risk and standard deviation was then calculated for each production system and for overall study area. Incidence risk of diarrhea and respiratory disease were also calculated for the previous 12 months by dividing the number of affected animals by the total number of live-born animals.

To evaluate statistical significance of the findings, binomially distributed generalized linear mixed effects modelling for risk of mortality, diarrhea, and respiratory disease was performed in R via RStudio. The lme4 and sjPlot packages were used to conduct and output the results of the analysis.

## 3. Results

### 3.1. Epidemiological Arm

#### 3.1.1. Household and Herd Characteristics

Household- and herd-level risk factor variables were grouped and analyzed. Results include data only from a subset of farms due to lack of data and/or data discrepancies collected from pastoral, mixed crop–livestock, and peri-urban farms (Table 3). In each case, this is specified.

#### 3.1.2. Owner Demographics

Across production systems, animals were predominantly owned by men, with mixed crop–livestock farms associated with the highest number of female livestock owners. The education level of farm owners was most commonly “None or preschool” in mixed crop–livestock farms (63.2% of owners), as well as in among pastoral herds (84.5%), whereas the most common education level among peri-urban farm owners was primary school (46.6%). Among livestock owners, very few had achieved a higher-level education, with the exception of 5.2% of peri-urban farm owners. For the majority of peri-urban farmers, farm produce was the main source of income, with no significant statistical differences between female-owned and male-owned farms (70% and 72.1% in female-owned and male-owned farms, respectively; *p* = 1.0). The majority of farms had been owned for five or more years.

#### 3.1.3. Herd Size

Average herd size and average number of calves varied by production system; pastoral herds had the largest average herd size (39.56 cattle/herd), followed by peri-urban herds (12.93 cattle/herd) and mixed crop–livestock herds (4.53 cattle /herd). The number of calves by production system followed the same pattern, with pastoral herds having the most calves, followed by peri-urban and mixed crop–livestock. As expected, there was an association between herd size and number of calves, with larger herds having a larger number of calves (*p* < 0.001). Proportionally, peri-urban farms had a higher number of adult female animals when comparing mean number of adult animals on a farm: 71.5% of animals on peri-urban farms were female, compared to 46.7% of pastoral herds and 50.6% of mixed crop–livestock farms. This intuitively makes sense, as the economically productive animals on a dairy farm are the lactating females. Local breeds were predominant in pastoral herds and mixed crop–livestock farms (90.9 and 91.3%, respectively), whereas only 17.1% of animals on peri-urban farms were local breeds, with Holstein being the predominant breed in this production system (89.7%). All calves from pastoral herds were Afar breed. Mixed crop–livestock and pastoral herds used predominantly live cover (96.1 and 100%, respectively), whereas 53.4% of peri-urban farms used artificial insemination.

#### 3.1.4. Dam-Level Variables

The distribution of dam parity, milk yield (liters), age at parturition (years), and body condition score (BCS, 1–5) [17] varied by production system. Pastoral herds tended to have cows with the greatest number of parities, followed by peri-urban and mixed crop–livestock farms, although the distribution of dam age at parturition did not quite follow the same pattern, and age distribution in pastoral herds was more distributed compared with peri-urban farms. Mixed crop–livestock cows were the thinnest, with a mean BCS of 2.21, followed by pastoral and peri-urban cows. As expected, milk yield was substantially increased on peri-urban farms compared to pastoral and mixed crop–livestock farms.

#### 3.1.5. Housing, Closeup Pens, and Calving Facilities

In peri-urban and pastoral farms, most calves were housed separately from the herd, whereas in contrast, almost all calves on mixed crop–livestock farms were housed with the dam. Regarding housing space and location with respect to the herd, most farms provided enough space for the calves to move and turn around when they were housed in groups. Housing type was not associated with education level.

With regard to calving areas, no farms reported having calving facilities, and only 29.7% and 25.6% of the farms reported having a separate birth area and close-up pen (pen for animals close to parturition), respectively. Having a separate birth area was associated with education level; farmers with primary or secondary education were more likely to have a separate birth area compared to farmers with none or preschool education (28.2%, 14.4%, and 7.7%, respectively; *p* < 0.001). Owner gender and separate birth area were not associated; however, having a close-up pen was associated with female-owned farms (43.3% female-owned vs. 21.2% male-owned farms; *p* = 0.028). Similarly, close-up pens were also associated with owner’s education level, with a higher proportion of farmers with at least a secondary education being more likely to have a close-up pen when compared to those with none, preschool, or primary education (34.8%, 21.1%, and 16.3%, respectively; *p*-value = 0.071). We also found that although having a close-up pen was not the most common practice, the proportion of farms with a close-up pen was always higher among female-owned farms compared to male-owned farms, regardless of education level (*p* = 0.07).

#### 3.1.6. Water, Flooring, and Cleanliness

Only 17.8% of the pastoral herders reported providing water to their calves, whereas 97.7% and 100% of the mixed crop and peri-urban farmers reported providing water to their calves, respectively (*p* < 0.001). No farms reported treating the water for animal consumption, and no associations were found with owner education level or gender. Among those who provided water, the majority did so once a day in mixed crop–livestock farms and pastoral herds, whereas the majority of peri-urban farms provided water twice a day. No pastoral herds provided water in independent water troughs, instead only using natural water sources, and only a small proportion did so in mixed crop–livestock and peri-urban farms.

Sex of the calf was not associated with frequency of water provision (*p* = 0.604) or the location where water was offered (*p* = 0.890). Regarding floor type and cleanliness, 70% of farms had concrete floor surfaces, followed by soil and stone in similar proportions, each 15%. A proportion of 94.8% of farms were classified as having clean feeding and watering areas, with no significant differences according to education level or gender.

#### 3.1.7. Bedding Use

The majority of farms did not use bedding materials for their calves. A comparison of the production systems showed that only peri-urban and mixed-crop farmers used bedding, whereas none of the pastoral herders reported using bedding. Among those farms using bedding material, the majority were classified as clean (100% and 92.7% clean bedding in mixed-crop and peri-urban farms, respectively). In contrast, among those farms not using bedding materials, a comparatively higher proportion were considered not clean. Mixed crop–livestock farms ranked the cleanest, followed by peri-urban and pastoral herds (85.9%, 77%, and 66.4% with reported clean bedding in mixed crop–livestock, peri-urban, and pastoral herds, respectively). In other words, farmers or herders not using bedding had a harder time keeping the surface clean where the calves rest (*p* < 0.001).

### 3.2. Epidemiological Testing

For the epidemiological arm, we enrolled and sampled a total of 3544 animals from 1005 farms over two years of sample collection and testing, representing the three primary livestock production systems in Ethiopia. In aggregate, the number of households enrolled and animals tested is substantial. However, due to differences in how data were recorded or provided by individual graduate students associated with the project, as well as financial limitations on the number of tests that could be performed per animal, test results are presented, and risk factor analyses were performed only on trusted and reliable datasets.

#### 3.2.1. Calf Physical Exam Information

Physical exam findings (Table 4) revealed significant associations between production system and rectal temperature score [18], as well as between BCS and fecal score, with more abnormal recordings in pastoral and peri-urban farms (*p* < 0.001). There was a higher incidence of thin animals in mixed crop–livestock farms compared to pastoral herds and peri-urban farms. There was an association between rectal temperature score and fecal score (higher fecal score = increased diarrhea severity and higher rectal temperature score; *p* = 0.009), higher fecal score was associated with lower BCS, and higher BCS was associated with lower fecal score; *p* < 0.001 (Table 5).

#### 3.2.2. Neonatal Diarrhea Complex

Calf diarrhea is a multifactorial disease caused by a host of pathogenic and non-pathogenic factors. The distribution of specimens tested with a Pathasure antigen ELISA kit by production system is as presented in Table 6. Complete Pathasure kit results are available for the pastoralist and peri-urban areas, whereas only *E. coli* K99 results are available for the mixed crop–livestock system.

In both pastoral and peri-urban production systems, *C. parvum* was the most common pathogen identified, followed by *E. coli* K99. When disaggregated by age group, *C. parvum* remained the most common pathogen identified in all age groups in both production systems. *E. coli* K99 infections were significantly higher in animals less than 1 month old (*p* = 0.007; Figure 2) which is expected based on existing literature and experience with dairy calves around the world. On peri-urban farms, younger animals were more likely to test positive for *C. parvum* and rotavirus. In contrast, animals >2 months old in pastoral herds were more likely to test positive for *C. parvum* and rotavirus. Animals older than 6 months (data not shown) were much more likely to test positive for bovine coronavirus; however, this age group was not the target age group for our study and was therefore a much smaller sample size than the other age groups analyzed.

#### 3.2.3. Fecal Parasites

Gastrointestinal parasite burden in calves can cause a variety of clinical syndromes, including diarrhea, poor thrift, and compromised immune function. As indicated in Table 7, a variety of fecal parasites were identified. The most commonly identified parasites in both production systems were *Strongyloides* spp. *Strongyloides* spp. infections were significantly more likely to occur in animals from mixed crop–livestock systems. Coccidia was often found in calves from peri-urban farms; however, it was not tested for in mixed crop–livestock samples.

#### 3.2.4. Respiratory Viruses and Bacteria

Respiratory disease in calves is often caused by a complex interaction of several co-infections with viral and bacterial pathogens and is often referred to as bovine respiratory disease complex (BRDC). Due to testing capacity and laboratory limitations, respiratory bacteria and viruses were only tested in Gondar (peri-urban farms), with a total of 275 animals tested for at least one respiratory pathogen (Table 8).

The most commonly identified respiratory pathogen in tested samples was bovine adenovirus (ADV), followed by parainfluenza virus-3 (PIV3) and bovine respiratory syncytial virus (BRSV), with a high prevalence of each of these three pathogens identified. The most common respiratory coinfection combinations included ADV + BRSV + PIV3 (*n* = 17, 19.8%), ADV+ BRSV + PIV3 + *M. haemolytica* (*n* = 15, 17.5%), and ADV + BRSV + PIV3 + IBR (*n* = 8, 9.4%). For bacterial infections, *Mannheimia hemolytica* was more likely to be cultured than *Pasteurella multocida.* When disaggregated by pathogen, there were no differences in the distribution of positive samples across age groups, parity, or dam age at parturition. 

#### 3.2.5. Colostrum, Passive Transfer, and Supplemental Feeding

All peri-urban producers reported being aware of the importance of colostrum feeding, with most calves receiving colostrum in the first six hours of life (76.9%). All calves received milk (vs. milk replacer), and most received it as residual suckling (95.5%) two times per day. Whereas the majority of farms reported that calves received colostrum within 24 h (96.6%, 100%, and 95.1% in mixed crop, pastoral, and peri-urban farms, respectively), the number of peri-urban calves that did not receive colostrum within 24 h was significantly less than in other production systems (*p* < 0.001). Access to their dam in the first 24 h also varied by production system (99.4%, 98.9%, and 89% in mixed crop, pastoral, and peri-urban farms, respectively), and similarly, a significantly higher number of calves did not have continuous access to the dam in the first 24 h in peri-urban farms compared with calves in mixed-crop and pastoral herds (*p* < 0.001).

Results for passive transfer of immunoglobulins (IgG, Table 9) were obtained in pastoral and peri-urban animals only, with the majority of results available for pastoral calves.

In the pastoral production system, 20.7% of calves showed evidence of partial or complete failure of passive transfer, whereas in peri-urban calves 28.6% showed evidence of partial or complete failure of passive transfer. There were no significant differences according to production system. Among those calves with partial or failure of passive transfer, a higher proportion of calves was born from cattle with lower parities (one or two versus three or more, *p* = 0.073, Table 10). Due to sample size limitations, distribution of immune passive transfer by production system and parity was only evaluated for pastoral calves. There were no statistical differences across parities with respect to failure of passive transfer (one, two, or three versus four or more, *p* = 0.419). However, partial/failure (to) transfer tended to decrease as dam parity increased.

Across all three production systems, the majority of pre-weaned calves were fed milk (95.8%, 99.9%, and 100% in mixed-crop, peri-urban farms, and pastoral herds, respectively). Only 0.8% of calves were fed milk replacer, and this practice only occurred on mixed crop–livestock farms. Calves should be fed 2 L twice a day; the vast majority of calves in this study were underfed in terms of both frequency and volume. The situation was severe on mixed-crop farms, where 44.3% of calves received less than a half-liter of milk per feeding; the findings in peri-urban and pastoral herds and peri-urban farms were similar, where 16.4 and 14.6% of calves received less than a half-liter per feeding, respectively (Table 11). Only 11.5% of calves in peri-urban farms received more than 1 L of milk per feeding. There were no differences in calf sex in terms of the amount of milk received (*p* = 0.237); however, there were differences based on the sire, with calves born from artificial insemination (AI) receiving more milk than those born from live cover (*p* < 0.001; data not shown). Supplemental feeds were provided to 50% of calves in mixed crop–livestock farms and to 86.0% of calves in peri-urban farms (*p* < 0.001; data not shown). Virtually no supplemental feed was provided in pastoral herds. In summary, the majority of calves in the study received inadequate amounts of milk and supplemental feeds and had inadequate nutrition.

### 3.3. Intervention Arm

Within the intervention arm, 856 households were enrolled in the study at baseline. During the final evaluation, 52 households were lost to follow-up (61.5% of losses from pastoralist, 27% from peri-urban, and 11.5% from mixed crop–livestock areas). In the process of data cleaning, households in which no calves were born in the previous year or where data appeared to contain errors were removed from the study. The results from 646 households are included in the subsequent statistical summaries and analysis (Table 12).

#### 3.3.1. Intervention Uptake

Figure 3, Figure 4 and Figure 5 show the percentage of households either with an optimal level of intervention practices from the start, having made an improvement in their practices, or with no or negative change in practices in the mixed crop–livestock, pastoral, and peri-urban production systems, respectively. The interventions are listed in order of positive change.

Intervention uptake differed between production systems. For the mixed crop–livestock system, navel dipping, examination of sick calves, and provision of supplementary feed to calves had the highest levels of uptake. For both pastoralists and peri-urban producers, the provision of supplementary feed to cows during pregnancy and separation of pregnant cows were the two most practiced interventions. This was followed by examination of sick calves in pastoralist areas and provision of calf supplementary feed for peri-urban calves.

#### 3.3.2. Mortality, Diarrhea, and Respiratory Disease Risk

Reduction in youngstock mortality and incidence of diarrhea and respiratory disease were objectives of the interventions. Table 13 shows the summary statistics at baseline and final evaluation for these parameters.

#### 3.3.3. Calf Mortality

Between baseline and final evaluations, the risk of calf mortality reduced in all production systems but most notably in the pastoralist system (Figure 6). These differences were analyzed using a binomial mixed-effects model fitted by maximum likelihood. Individual farms and geographical area (districts) were treated as nested clusters and included in the model as random effects, whereas evaluation round (baseline or final) and production system (mixed crop–livestock, pastoral, or peri-urban) were tested as fixed effects.

Overall, the odds ratio (OR) of mortality occurring in the previous 12 months at final evaluation compared to baseline was 0.35 (*p* < 0.001) (Table 14). The difference in mortality risk between the mixed crop–livestock and peri-urban systems was not significant (*p* = 0.067); however, risk of mortality for calves raised in the pastoral system was significantly higher: 3.43 times the risk compared to the mixed crop–livestock system (*p* = 0.012).

#### 3.3.4. Calf Diarrhea

The average risk of calf diarrhea reduced in all production systems at final compared to baseline evaluation, again with the greatest reduction observed in the pastoralist system (Figure 7).

Calf diarrhea was modelled in the same way as calf mortality. Overall, the OR of calves experiencing diarrhea was 0.2 compared to baseline (*p* < 0.001) (Table 15). Calves raised in the pastoral system were predicted to have approximately double the odds of having diarrhea compared to mixed crop–livestock calves, but this finding had borderline significance (*p* = 0.042). The difference in calf diarrhea between mixed crop–livestock and peri-urban systems was not significant.

#### 3.3.5. Calf Respiratory Disease

The average risk of respiratory disease in calves reduced in all production systems at final compared to baseline evaluation; however, the incidence of respiratory disease was much lower than that of diarrhea (Figure 8). The mixed-effects modelling of respiratory disease risk shows that the OR of calves with respiratory disease at final evaluation was 0.29 compared to baseline (*p* < 0.001). There were no significant differences between the three production systems (Table 16).

## 4. Discussion

The results and data presented in this study provide valuable information on livestock production practices, risk factors, and respiratory and diarrheal disease pathogens related to calf morbidity and mortality in Ethiopia. Broad in scope and ambition, the YSMRC generated important insights that can inform livestock extension agents and peri-urban, mixed crop–livestock, and pastoral producers on ways to improve calf health and overall herd production.

Within the epidemiological arm, the study generated valuable new information regarding the prevalence of specific pathogens that cause diarrhea and respiratory disease in calves. Significant associations between production system and rectal temperature score, body condition score, and fecal score were observed, with the association between fecal score and BCS showing an opposite trend. These findings make sense clinically, in that animals with clinical signs of diarrhea were more likely to show evidence of a fever and/or be in worse physical condition. In both pastoral and peri-urban production systems, *C. parvum* was the most common pathogen, followed by *E. Coli* K99. Asmare and Kiros (2016) [8] also identified *Cryptosporidium* as the most prevalent pathogen (52.6%) in diarrheic dairy calves, and Ayele et al. (2018) [19] identified *Cryptosporidium* in 18.6% of mixed farm calves. More recent studies observed *Cryptosporidium* in 13.8% of calves in a mixed crop–livestock system [20], with calves under intensive management systems more likely to be infected than calves in extensive systems [21]. Our findings for *C. parvum* in different age groups were slightly unexpected, as *C. parvum* most commonly appears in older animals in more developed economies, once animals are kept together in pens. However, this is consistent with the results reported by Asmare and Kiros (2016) [8], who found *Cryptosporidium* in 10 of 19 calves <6 months. Because most of the young stock enrolled in the YSM study were cohoused with animals of all ages, it is possible that is why younger animals on peri-urban farms were as likely to test positive as older animals (>6 mo.). The presence of bovine CoV in younger cattle suggests that it is present on farms, and the longer calves are on a farm, the more likely they may be to be exposed to the virus. The proportion of calves testing positive to CoV in this study is higher than previously reported; Seid et al. (2020) found CoV in only 1 of 83 diarrheic calves [22]. Bovine CoV can be an important pathogen on farms affecting calves and adults and deserves more investigation. The Pathasure Enteritis 4 diagnostic kit proved a useful and affordable diagnostic tool to employ in the field; additional studies that focus specifically on animals clinically affected by diarrhea paired with more detailed physical exam data could yield more insight into the variations across production systems and inform future preventive care strategies, including vaccination.

*C. parvum* is a zoonotic pathogen capable of causing diarrhea in children and adults and is associated with environmental enteropathy, malnutrition, and stunting in young children in settings with poor water, sanitation, and hygiene [23,24]. Given the proximity of human and livestock housing on some farms in Ethiopia and the impact improved hygiene could have on both calf and human health, an increased focus on common zoonotic pathogens and their relationship with environmental enteropathy is an area worthy of substantially more scientific inquiry. Applying advanced molecular diagnostic techniques in tandem with concurrent sampling of livestock, children, other household members, other domestic animals, and environmental samples in households is an approach that has the potential to yield important information about how infections are spread (i.e., from animals to humans or between humans), the importance of coinfections and comorbidities, and the importance of symptomatic vs. asymptomatic cases with regard to pathogen shedding.

A variety of fecal parasites were identified, with the most common in both mixed crop–livestock and peri-urban farms being *Strongyloides* spp. and significantly more likely in mixed crop–livestock systems. Within the epidemiological arm, only 1% of peri-urban producers and no mixed crop–livestock producers reported regular use of anthelmintics in young calves; however, anecdotal experience of several authors of this paper indicate that regular deworming of yearling and adult cattle is common. There are likely several reasons that account for the difference in prevalence rates, including differences in deworming practices, housing, and/or grazing practices in the different production systems, as well as potential anthelmintic resistance. Future intervention packages could consider adopting targeted deworming strategies informed by clinical examination and fecal egg count reduction tests [25].

The most commonly identified respiratory pathogens in the tested serum samples tested were bovine adenovirus, parainfluenza virus-3 (PIV-3), and bovine respiratory syncytial virus (BRSV). We found no differences in respiratory infection prevalence by age group in peri-urban calves (due to several limitations, in the epidemiological arm, we were only able to collect respiratory pathogen seroprevalence data in peri-urban calves). Viral respiratory pathogens in Ethiopian calves are generally poorly described in the literature. A comparison across a wider geography identified PIV-3 in 20.1% of calves in mixed crop–livestock systems in Kenya [26], whereas in a recent Belgian study of dairy, dairy-mixed, and beef calves, BRSV was the most commonly isolated respiratory pathogen (29.4%), and PIV-3 was isolated in 8.1% of outbreaks [27]. A study of Finnish dairy calves reported BRSV in 19% (serum samples) and PIV-3 in 16% of calves [28]. The majority of the respiratory disease surveillance data relied on serological assays, which can be complicated to interpret in young calves <3 months of age due to maternal antibody interference. This means that a positive test in a young calf may mean the dam was either previously infected (or vaccinated, which is extremely unlikely in this context) and then passed on antibodies to the calf when the calf consumed colostrum or that the calf produced its own antibodies in response to a naturally occurring infection. Because the herders whose animals were sampled do not regularly vaccinate their dams for any respiratory pathogens, a positive test result indicates that, at the very least, these pathogens are circulating in the herd and have the potential to cause disease in both young and adult animals. As calves age, maternal antibodies wane, so the lack of a decrease in prevalence in the older calves supports the fact that calves were infected by these common respiratory pathogens on peri-urban farms in Gondar. Future research to further explore respiratory infections would require longitudinal sampling and paired titers the use of nucleic acid testing assays, such as reverse transcriptase polymerase chain reaction (RT-PCR), for confirmation of infection and/or pathogen identification.

In the intervention arm, baseline levels of mortality in the peri-urban system were similar to those reported in recent studies by Romha (2014) [7], Asmare and Kiros (2016) [8], and Fentie et al. (2020) [9] but lower in the mixed crop–livestock group compared to the results reported by Ferede et al. (2014) [10]. As delivered, the targeted interventions were shown to definitively decrease calf mortality and reduce the incidence of diarrhea and respiratory disease in calves. Overall, households enrolled in the study showed significant reductions in incidence risk of mortality, diarrhea and respiratory disease. All production systems showed reduced mortality, with 64.2%, 72.4%, and 31.4% reductions for mixed crop–livestock, pastoralist, and peri-urban systems, respectively.

In the intervention arm, overall (i.e., for all systems in all areas), the interventions that had the greatest level of uptake were the provision of supplementary feed during pregnancy and for calves, navel dipping of calves, and separating pregnant cows. As stated, provision of colostrum, use of calf pens, and examination of sick calves were already reported to be practiced by a high percent of households at baseline. Within each production system, different interventions were taken up; producers in the mixed crop–livestock system demonstrated greater uptake of health and hygiene interventions, whereas pastoralist and peri-urban producers had greater uptake for interventions providing nutritional support to pregnant cows and calves. The uptake of interventions may reflect pre-existing knowledge of farmers; accessibility of inputs, including those provided through YSMRC funding; what farmers consider most important; or what was most efficient for farmers to incorporate into their current practices. A repeat evaluation to investigate the sustainability of interventions after conclusion of the YSMRC support and assess long-term impact, as well as further research as to why some interventions were more successful than others, may support the refinement of future intervention packages.

In both the epidemiological intervention arms, the need for adequate nutrition for dams and calves was highlighted. In particular, the importance of adequate colostrum intake within the first few hours after birth cannot be overstated. Other studies in Ethiopia have reported similar findings, with higher mortality rates associated with inadequate colostrum and delayed colostrum [7,8,29]. Neonatal ruminants are dependent on the ingestion of colostrum to obtain crucial antibodies from the dam that support their immune system. Failure of passive transfer of colostral antibodies means that a calf is more susceptible to infectious disease early in life and is a known risk factor for future morbidity and mortality, particularly from respiratory and diarrheal diseases. Many of the enrolled farms in the Intervention arm of the study reported high levels of practicing colostrum management. Despite a high reported level of feeding colostrum, 20.7% of pastoral and 28.6% peri-urban calves exhibited partial or complete failure of passive antibody transfer. This difference highlights an important nuance. Just because a farmer reports a practice does not always mean they are effective at or knowledgeable about the practice, and ensuring a calf suckles after birth does not always translate into a calf consuming enough high-quality colostrum from the dam. Importantly, in this study, we also identified that in general, calves and cows did not receive enough milk or supplemental food in the evaluated population. Underfeeding, particularly of milk in the first few weeks of a calf’s life, likely contributes to poor health outcomes. Market and household economic pressures to sell milk (rather than feeding it to calves), unavailability or prohibitively high cost of milk replacer and other supplemental feeds, farmer knowledge of how much milk a young calf requires, and other factors are all possible causes of the underfeeding observed in this study. However, after receiving training, farmers often readily adopted supplemental feeding as a new practice. Identifying misconceptions, barriers, and facilitators of both colostrum and supplemental feeding practices warrants further investigation.

This study is subject to several limitations and lessons learned. Due to a series of implementation challenges, although there was substantial overlap of the households in the four regions where both the epidemiological and intervention arms were conducted, we were not able to aggregate household or animal data across both arms. Furthermore, a large number of households were lost to follow-up or during the data-cleaning process in both arms of the study.

Within the intervention arm, more pastoralist participants (61.5% of households lost to follow-up) did not participate in the final evaluation compared to mixed-crop (11.5%) and peri-urban households (27%). The difficulties of enrolling pastoralist households included similarities in names across households and the lack of fixed addresses to definitively identify households or to follow-up. The difficulties in reaching pastoralist communities likely contribute to both the dearth of data available and the level of support pastoralists are able to access and is an area that deserves more dedicated attention and research. Additionally, many households in the mixed crop–livestock production system were excluded from the intervention-arm analysis due to no calves being born in the 12 months prior to the baseline or final evaluations. A larger sample size and/or longer research period is recommended for future studies, and reproductive management should be considered as an area for farmer training. Finally, the number of data entries censured for incompleteness, inaccuracy, or implausibility could be improved by enhanced enumerator training, checks and balances built into the questionnaires, or with the use of electronic data collection systems that can automatically flag missing or aberrant data, whereas smaller degrees of change could be captured with improved questionnaire response options.

Another important limitation of the intervention arm arose through the quantity of data it was feasible to collect, as well as the use of a package of interventions, which, combined, did not allow the analysis to identify the intervention(s) that contributed most significantly to change. Understanding the contribution of individual practices could be useful in terms of prioritizing interventions and analyzing the return on investment for producers, particularly when maintaining certain practices requires significant producer or government expenditure.

We also experienced challenges in the epidemiological arm with respect to data depth, quality, and consistency. Due to financial and logistical constraints, it was not possible to test each animal enrolled for the full range of pathogens. Procuring the appropriate testing supplies, kits, and laboratory reagents in Ethiopia was incredibly time- and resource-intensive. A portion of the funding was focused on capacity development, and Ethiopian graduate and veterinary students were trained to assist in data collection and laboratory testing on this project. Variability in the level of student engagement, competency, and proficiency in animal handling and physical examination, as well as attention to detail (especially with record keeping and data entry) and faculty oversight, all contributed to variations in the quality and consistency of data collected in the epidemiological arm. The study team worked to address these challenges by providing standardized data collection protocols, several in-person trainings, and remote mentoring with the students that worked with the project. A useful lesson was that placing students together in groups of three or four dramatically improved the quality of data collected in the field, as they could work together while performing household surveys and clinical exams, collecting specimens, and performing laboratory diagnostics.

Despite the limitations and challenges encountered, we believe that the information gleaned through this effort is useful for Ethiopian farmers, livestock extension workers, and policy makers. Additionally, the working relationships of the YSMRC formed an invaluable foundation for future collaborative work and set a new precedent for a collaborative funding model for livestock development work in Ethiopia. Throughout the study period, the YSMRC was committed to developing human and institutional capacity through dedicated trainings, mentorship, and oversight, working with faculty professors, graduate students, and laboratory staff, along with extension officers and livestock keepers, through the YSMRC intervention activities. Specifically, 28 students from 7 universities. including 15 veterinary students and 13 graduate students, were supported by the project. All gained valuable field and laboratory experience to supplement their education and, importantly, helped contribute to the national food security, nutrition, and economic development objectives of the Ethiopian veterinary and livestock research sectors.

At the time of publication, we are aware that the activities piloted under the intervention arm are currently being scaled-up by the Ethiopian government based on the findings and recommendations of the YSMRC study. The epidemiological arm revealed that certain behaviors and practices associated with improved health outcomes on farms were associated with education, gender, education, use of AI for breeding, and/or production system. This included use of a separate birthing area, use of a close-up pen, use of bedding, record keeping, feeding practices, and quarantining new animals. Several reasons or explanations for this may exist, and confounders, such as household income and training or supervision received by private AI technicians, may be responsible for some of these associations. However, as the intervention packages are scaled by the MoA, recognizing the importance of prior or repetitive training, making packages gender-sensitive, identifying practices most likely to be adopted by producers, and considering which require the most external support to be sustained are all important points to consider. Training for farmers on how to identify at-risk calves and how to prioritize treatment of the sickest animals may represent a second level of training that could be delivered. Furthermore, improving the understanding of the nuanced differences that affect behaviors across the different production systems, particularly those surrounding gender and culture, might be especially important to refine the intervention packages and overcome long-held beliefs and practices associated with livestock ownership. Additionally, interventions could be evaluated from a perspective of what is most economically worthwhile while simultaneously evaluating the time benefit and tradeoff in time and effort at the household level, particularly though a gender lens.

## 5. Conclusions

Our findings support what experienced livestock practitioners and successful farmers likely already know; attentive care with respect to how cows and calves are raised and housed, keeping them on dry and clean bedding and housing, ensuring adequate and timely feeding of quality colostrum, and ensuring the animals receive adequate, high-quality food and clean water are fundamental steps in maximizing animal health and wellbeing. The degree of reduction in youngstock mortality achieved by the interventions piloted in this study highlights the importance of adequate husbandry and management. In the international development arena, much of the recent attention on animal health has been focused on the control of transboundary animal and zoonotic diseases, which have important trade and public health implications. However, for the majority of small-scale producers, the impact of persistent poor nutrition, neonatal management, hygiene and sanitation, and availability of veterinary care is far greater, and interventions targeting these issues require more international support, particularly among pastoralist producers. Continued investment in targeted strategies designed to improve producer knowledge and adoption of basic improvements in animal husbandry, housing, and feeding practices are first line-interventions recommended to reduce young stock morbidity and mortality in Ethiopia.

## Figures and Tables

**Figure 1 animals-12-02126-f001:**
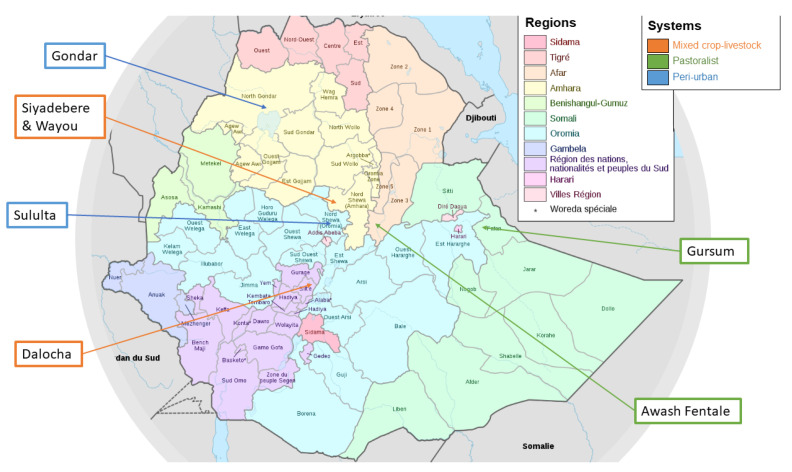
Map of Ethiopia with study sites labelled. Map by Bouzinc, 2020—Own work, CC BY-SA 4.0, https://commons.wikimedia.org/w/index.php?curid=95981571 (accessed on 10 August 2021).

**Figure 2 animals-12-02126-f002:**
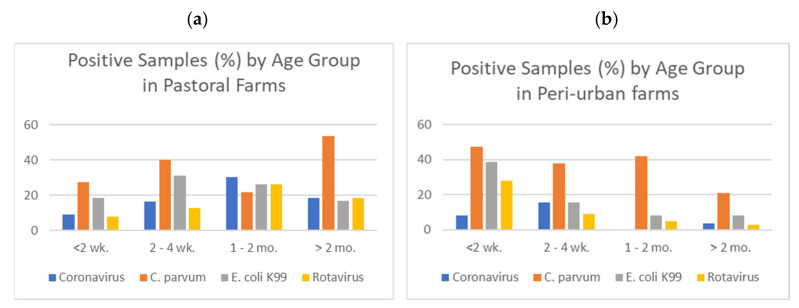
Diarrhea test results by age group in (**a**) pastoral and (**b**) peri-urban production systems.

**Figure 3 animals-12-02126-f003:**
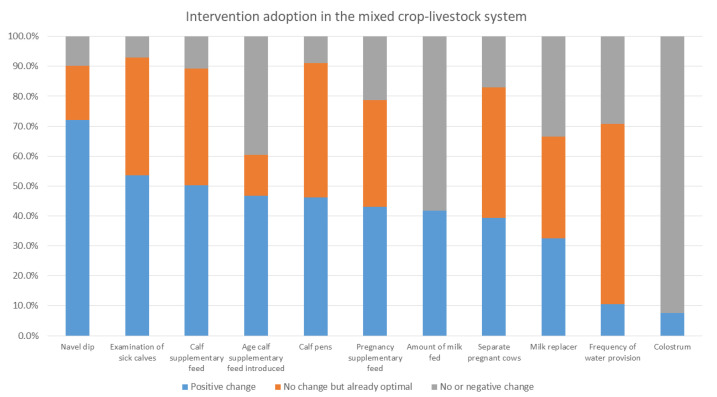
The proportion of enrolled mixed crop–livestock households with baseline and final evaluation data with either (a) no change in intervention area practices during the study period but were already optimal at the start of the study, (b) positive change, or (c) no or negative change in intervention areas.

**Figure 4 animals-12-02126-f004:**
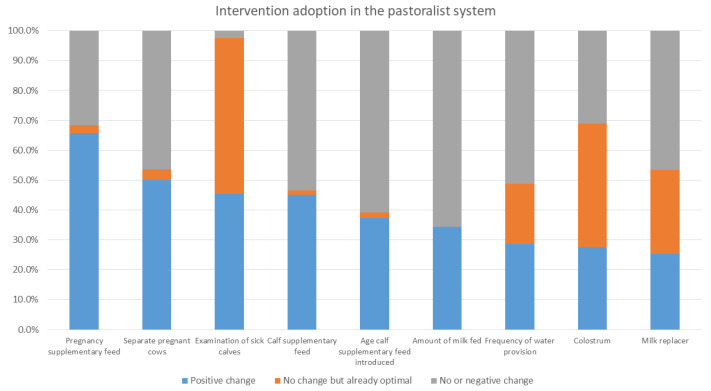
The proportion of enrolled pastoralist households with baseline and final evaluation data with either (a) no change in intervention area practices during the study period but were already optimal at the start of the study, (b) positive change, or (c) no or negative change in intervention areas. N.B. Navel dipping and calf pens were not selected interventions for the pastoral system.

**Figure 5 animals-12-02126-f005:**
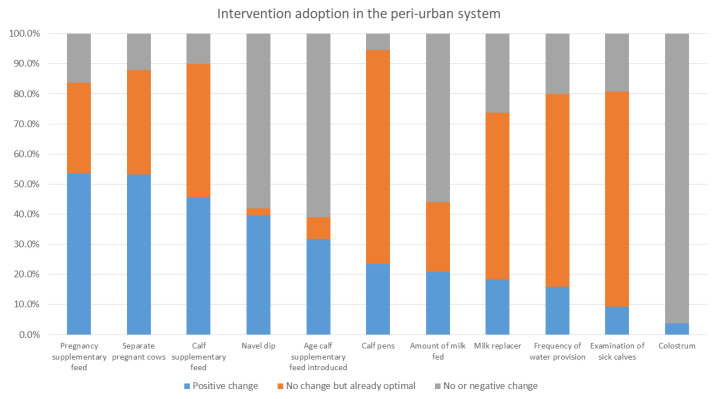
The proportion of enrolled peri-urban households with baseline and final evaluation data with either (a) no change in intervention area practices during the study period but were already optimal at the start of the study, (b) positive change, or (c) no or negative change in intervention areas.

**Figure 6 animals-12-02126-f006:**
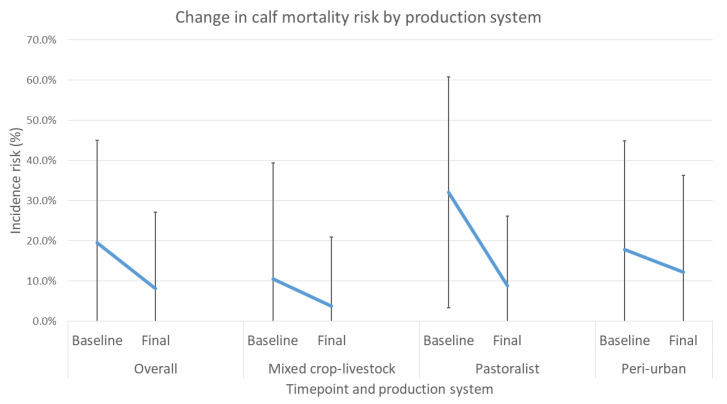
Change in average mortality risk in calves between baseline and final overall evaluation and for each production system with standard deviation bars.

**Figure 7 animals-12-02126-f007:**
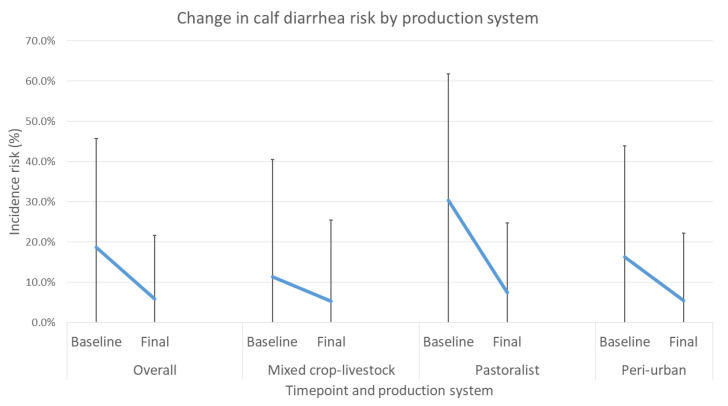
Change in average diarrhea risk in calves between baseline and final overall evaluation and for each production system with standard deviation bars.

**Figure 8 animals-12-02126-f008:**
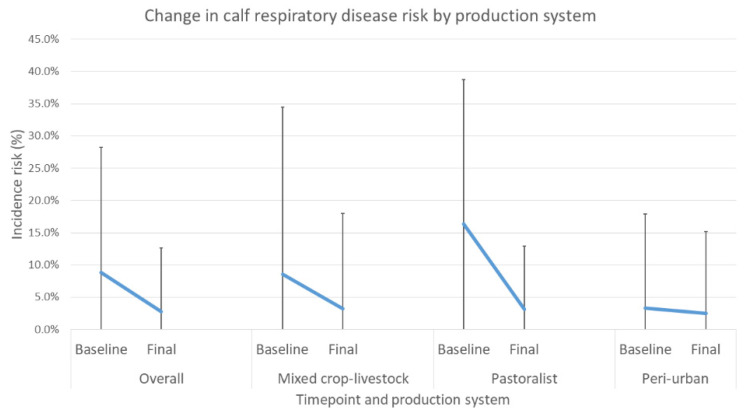
Change in average respiratory disease risk in calves between baseline and final overall evaluation and for each production system with standard deviation bars.

**Table 1 animals-12-02126-t001:** Region, district, and production systems of the six study areas *.

Region	Woreda/District *	Production System
Afar	Awash-Fentale	Pastoralist
Somali	Gursum	Pastoralist
Oromia	Sululta	Peri-urban
Amhara	Gondar	Peri-urban
Amhara	Siyadabere and Wayou	Mixed crop–livestock
SNNP	Dalocha	Mixed crop–livestock

* The epidemiological arm included Awash-Fentale, Suluta, Gondar, and Dalocha; the intervention arm included all districts.

**Table 2 animals-12-02126-t002:** Name and description of interventions for which household-level data were collected at baseline and final evaluations.

Intervention	Description of Recommended Practice	Question Asked
Pregnancy supplementary feed	Supplementing dam feed during the last trimester with locally available feeds, urea treated straw, concentrate, legumes, or fortified lick.	Do you provide feed supplements for pregnant cows near to parturition?
Navel dip *	Tie and cut umbilicus and dip stump in antiseptic solution or apply antibiotic spray.	Did you dip the navel of newborn calves in iodine immediately after birth?
Separate pregnant cows	Separating dams around the time of parturition and performing regular inspections.	Do you keep pregnant cows separated during parturition?
Calf supplementary feed	Provide hay, water, and protein supplement or calf starter from 3 weeks of age.	Do you provide supplementary feed (other than milk or milk replacement) to non-weaned calves?
Age calf supplementary feed introduced	Introduce calf starter feed at 21 days of age.	When do you introduce supplementary feed different from milk/milk replacer to calves?
Amount of milk fed	Ensure dam producing sufficient milk and allow calf to suckle at least one quarter from 5 to 21 days of age.	What is the amount of milk fed daily to newborn calves?
Examination of sick calves	Seek help from animal health professionals when calves are sick to enable appropriate treatment and sample collection.	Are sick calves examined for disease by health personnel?
Calf pens *	Where used, ensure appropriate flooring and bedding	Do you have separate calf pens?
Milk replacer	Implement foster/nipple/bucket feeding if dam is not producing sufficient milk.	Do you provide milk replacement to newborn calves?
Frequency of water provision	Ensure calves are offered fresh water ad libitum.	How often do you provide water to non-weaned calves?
Colostrum	Check colostrum production and ensure newborn suckles dam within 2 or at least first 6 h; ensure newborn imbibes an adequate volume of colostrum at first feed and over first 4 days.	Did the calves born during the last year get colostrum in the first day of life?

* Interventions not considered appropriate for the pastoralist system and not included in the monitoring framework for this system.

**Table 3 animals-12-02126-t003:** Household demographics, herd sizes, and production practices for households enrolled in the epidemiological arm.

	Production System			
	Mixed Crop–Livestock	Pastoral	Peri-Urban	*p*
Owner Demographics				
*n*	260	110	493	
Owner gender (female, %)	83 (31.7)	16 (14.5)	97 (16.5)	<0.001
Education level				<0.001
None or preschool	163 (63.2)	93 (84.5)	104 (23.5)	
Primary	68 (26.4)	15 (13.6)	206 (46.6)	
Secondary	26 (10.1)	2 (1.8)	109 (24.7)	
Higher	1 (0.4)	0 (0.0)	23 (5.2)	
Herd Size				
*n*	260	110	493	
Number of calves (mean, SD)	1.22 (0.59)	4.89 (3.89)	2.85 (2.18)	<0.001
Number adult females (mean, SD)	2.29 (1.31)	18.48 (12.31)	9.24 (11.25)	<0.001
Herd size (mean, SD)	4.53 (2.12)	39.56 (35.28)	12.93 (12.53)	<0.001
Dam Descriptive Details				
*n*	358	441	681	
Parity (mean (SD))	2.73 (1.38)	3.42 (2.36)	2.78 (1.44)	<0.001
Milk yield (mean (SD))	1.49 (1.90)	1.87 (2.03)	7.60 (4.91)	<0.001
Age at parturition (mean (SD))	6.82 (2.09)	6.45 (2.59)	7.06 (2.30)	0.001
BCS (mean (SD))	2.21 (0.62)	2.89 (0.65)	2.99 (0.78)	<0.001
Calf Housing				
*n*	358	441	681	
Group housing, enough space (yes, %)	345 (96.9)	391 (88.9)	325 (82.5)	<0.001
Calf housed with dam (yes, %)	355 (99.7)	4 (0.9)	39 (9.9)	<0.001
Calf housed with other livestock (yes, %)	356 (100.0)	355 (80.7)	194 (49.2)	<0.001
Calf housed separate from herd (yes, %)	3 (0.8)	437 (99.3)	395 (82.8)	<0.001
Calf Watering				
*n* (%)	347	51	383	
Frequency of water provision				<0.001
Once a day	318 (91.6)	29 (56.9)	47 (12.3)	
Twice a day	23 (6.6)	20 (39.2)	297 (77.5)	
More than twice a day	6 (1.7)	2 (3.9)	39 (10.2)	
Water provided through independent water trough (yes)	96 (27.7)	0 (0.0)	55 (14.4)	<0.001

**Table 4 animals-12-02126-t004:** Calf physical exam findings.

	Production System			
	Mixed Crop–Livs.	Pastoral	Peri-Urban	*p*
*n* (%)	340	441	383	
Body condition score (1–5)				<0.001
1	79 (23.5)	41 (9.3)	74 (19.3)	
2	181 (53.9)	92 (21.0)	145 (37.9)	
3	63 (18.8)	235 (53.5)	145 (37.9)	
4+	13 (3.9)	71 (16.2)	19 (5.0)	
Rectal temp (≥38.9 °C)	47 (20.5)	112 (39.2)	128 (34.5)	<0.001
Fecal score (≤2)	13 (3.8)	47 (10.7)	151 (39.4)	<0.001

**Table 5 animals-12-02126-t005:** Fecal score and body condition score.

	Fecal Score *		
	≤1	≥2	*p*
*n* (%)	953	211	
BCS * (%)			<0.001
1	150 (15.8)	44 (20.9)	
2	308 (32.5)	110 (52.1)	
3	392 (41.4)	51 (24.2)	
4+	97 (10.2)	6 (2.8)	

* Fecal scoring and body condition scoring were determined in calves <6 months of age using protocols developed by animal health institutions [17,18].

**Table 6 animals-12-02126-t006:** Neonatal diarrhea complex results ^.

	Production System			
	Mixed Crop–Livestock	Pastoral	Peri-Urban	*p*
*n* (%)	40 (100%)	199 (100%)	281 (100%)	
Bovine coronavirus	---	32 (16.1)	15 (5.3)	<0.001 *
*Cryptosporidium parvum*	---	74 (37.2)	89 (31.7)	0.247 *
*Escherichia coli* K99	9 (22.5)	44 (22.1)	37 (13.2)	0.026
Bovine rotavirus	---	28 (14.1)	21 (7.5)	0.028 *

^ Pathasure antigen ELISA kits were used to test calves <6 months of age for neonatal diarrhea complex pathogens. * *p*-values for the association between pastoral and peri-urban production systems. --- these tests were not performed on samples in this production system.

**Table 7 animals-12-02126-t007:** Fecal parasite test results *.

	Mixed Crop–Livestock	Peri-Urban	
*n* (%)	192 (100%)	352 (100%)	*p*
Coccidia		123 (43.5)	
*Ascaris*	47 (25)	23 (21.7)	0.620
*Fasciola hepatica*	6 (3.1)	3 (4.3)	0.926
*Monezia* spp.	14 (7.4)	3 (4.3)	0.559
*Paramphistomum* spp.	3 (1.6)	0 (0.0)	0.694
*Schistosoma* spp.		1 (1.4)	
*Stronglyloides* spp.	96 (50.3)	28 (26.4)	<0.001
*Trichuris* spp.	7 (3.6)	6 (5.7)	0.604

* Traditional fecal flotation technique and microscopic examination were performed in calves <6 months of age.

**Table 8 animals-12-02126-t008:** Respiratory virus test results *.

	Peri-Urban (*n*, %)
Virus	
Bovine adenovirus (ADV)	137 (87.%)
Parainfluenza virus-3 (PIV3)	129 (82.7%)
Bovine respiratory syncytial virus (BRSV)	109 (69.9%)
Bovine herpes virus (BHV-1/IBR) *	
Positive	83 (30.9%)
Suspected	5 (1.9%)
Bacteria	
*Mannheimia haemolytica*	52 (35.1%)
*Pasteurella multocida*	19 (12.8%)

* A combination of diagnostic assays, including IDEXX serological assays, trivalent Ab test, BHV/IBR gB X3 Ab test, traditional microbiological bacterial culture, and sensitivity testing, were used in calves <6 months of age.

**Table 9 animals-12-02126-t009:** Passive transfer of immunoglobulins (IgG).

	**Production System**		
	Pastoral	Peri-Urban	*p*
*n* (%)	217	14	
IgG			0.503 *
Adequate transfer	172 (79.3)	10 (71.4)	
Partial/failure (to) transfer	45 (20.7)	4 (28.6)	

* Fisher’s exact test.

**Table 10 animals-12-02126-t010:** Immunoglobulins (IgG) by dam parity in pastoral calves.

	Parity				
	1	2	3	4+	*p*
*n* (%)	53	45	39	89	
IgG					0.419
Adequate transfer	40 (75.5)	33 (73.3)	30 (76.9)	75 (84.3)	
Partial/failure (to) transfer	13 (24.5)	12 (26.7)	9 (23.1)	14 (15.7)	

**Table 11 animals-12-02126-t011:** Frequency and volume of milk fed by production system.

			Amount of Milk or Milk Replacer		
	*n*	Frequency	Less than Half L	Half to 1 L	More than 1 L
Peri-Urban	262	Once a day	4	9	2
		Twice a day	27	141	26
		More than twice a day	12	39	2
Mixed crop–livestock	169	Once a day	30	42	0
		Twice a day	45	52	0
Pastoralist	41	Frequency data not available	6	34	1

Due to missing data, *p*-values were calculated separately. Milk volume fed vs. production system, *p* < 0.001; feeding frequency vs. amount of milk fed, *p =* 0.019.

**Table 12 animals-12-02126-t012:** Intervention-arm households enrolled at baseline, participating in the final evaluation, and remaining after data cleaning.

	Baseline	Final Evaluation	After Data Cleaning
Mixed crop–livestock	285	279	202
Pastoralist	271	239	204
Peri-urban	300	286	240
Total	**856**	**804**	**646**

**Table 13 animals-12-02126-t013:** Summary statistics for risk of mortality, diarrhea, and respiratory disease risk in calves.

Data Type	Baseline		Final		Change between Baseline and Final (%)	Change as a Percent of Baseline (%)
	Mean (%)	SD (%)	Mean (%)	SD (%)		
Mortality risk ^a^						
Overall	19.5	25.6	8.2	19.0	−11.4	−58.2
Mixed crop–livestock	10.5	28.9	3.8	17.2	−6.7	−64.2
Pastoralist	32.1	28.7	8.9	17.2	−23.2	−72.4
Peri-urban	17.9	27.0	12.3	24.1	−5.6	−31.4
Risk of diarrhea ^b^						
Overall	18.7	27.1	5.9	15.8	−12.8	−68.3
Mixed crop–livestock	11.4	29.2	5.4	20.1	−6.0	−52.6
Pastoralist	30.4	31.4	7.5	17.2	−22.9	−75.3
Peri-urban	16.3	27.5	5.5	16.8	−10.8	−66.2
Risk of respiratory disease ^c^						
Overall	8.9	19.3	2.8	9.9	−6.1	−68.3
Mixed crop–livestock	8.6	25.8	3.2	14.8	−5.4	−62.7
Pastoralist	16.3	22.4	3.1	9.8	−13.2	−80.8
Peri-urban	3.3	14.5	2.5	12.7	−0.8	−23.6

SD = standard deviation. ^a^ Calculated as total number of calves born alive but died/total number of calves born alive. ^b^ Calculated as total number of calves with diarrhea/total number of calves born alive. ^c^ Calculated as total number of calves with respiratory disease/total number of calves born alive.

**Table 14 animals-12-02126-t014:** Calf mortality model predictions.

Predictor	Level	Odds Ratio	95% Confidence Interval	*p*
Round	Baseline	1.00	-	-
	Final	0.35	0.29–0.41	<0.001
Production system	Mixed crop–livestock	1.00	-	-
	Pastoral	3.43	1.31–9.00	0.012
	Peri-urban	2.46	0.94–6.45	0.067

**Table 15 animals-12-02126-t015:** Calf diarrhea risk model predictions.

Predictor	Level	Odds Ratio	95% Confidence Interval	*p*
Round	Baseline	1.00	-	-
	Final	0.20	0.16–0.25	<0.001
Production system	Mixed crop–livestock	1.00	-	-
	Pastoral	2.48	1.04–5.93	0.042
	Peri-urban	1.39	0.58–3.33	0.463

**Table 16 animals-12-02126-t016:** Calf respiratory disease risk model predictions.

Predictor	Level	Odds Ratio	95% Confidence Interval	*p*
Round	Baseline	1.00	-	-
	Final	0.29	0.22–0.40	<0.001
Production system	Mixed crop–livestock	1.00	-	-
	Pastoral	1.96	0.53–7.24	0.314
	Peri-urban	0.34	0.09–1.3	0.116

## Data Availability

The datasets generated and analyzed during the current study are available at: epidemiological arm: https://dataverse.harvard.edu/dataverse/livestock-lab-ethiopia-young-stock-mortality (uploaded on 25 May 2022); intervention arm: https://doi.org/10.7910/DVN/NP0ABE (uploaded on 27 May 2022).

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
