# Peer review of "Reducing Calf Mortality in Ethiopia"

_animals, 2022, doi:10.3390/ani12162126_

Round 1
Reviewer 1 Report
An interesting study providing interesting demographics and baseline data
Simple Summary: "Over the last 20 years, studies carried out in Ethiopia report death and disease incidence rates in young livestock as high as 66%." this needs clarified as your introduction stated morbidity only up to 67% whereas mortality up to 20%
Abstract: Abstract should be re-written to discuss main findings, there is a lot of focus on background
Introduction:
There is a lot of focus on the funding which is in the funding section, could you just state the YSMRC was formed and the funding details in the funding section of the paper? This would need expanding but saves repetition
Materials and Methods:
L128-L131 not that clear what you mean consider making this clearer
otherwise this section is clear
Results:
Lot of information, do you need Figure 6 as the percentage change given in Table 13
Discussion:
The discussion is detailed and does cover a lot of the findings but does lack comparison with literature throughout, this could be improved
Reviewer 2 Report
This research reflects a task force of public entities and universities to collect epidemiological data and make interventions to decrease calves morbidity and mortality in the production systems from small herds located in different regions of Ethiopia. This is exemplary research that deserves to be published in full, in order to be an example for other institutions from developing countries. Regarding the content of the paper, I only recommend the review of all figures and tables, which are not self-explanatory in relation to the age of the sampled animals and the addition of the diagnosis principle (example tables 5, 6, 7 and 8). I also recommend the addition of referencesto endorse the criteria and cut-off points used to define fever and BCS, for example. I congratulate all the authors and authorities involved in the research, which certainly demonstrates the role of universities in the community.
Reviewer 3 Report
In atachment
